# Foundation for Chinese Poetry Research: An Open Large-Scale and Fine-Grained Multimodal Knowledge Graph

## Abstract

Classical Chinese poetry is a treasured cultural heritage of humanity, attracting extensive research interest. However, the study of classical Chinese poetry is hindered by the lack of open, large-scale, and fine-grained multimodal datasets. Prior datasets are either limited by modality constraints, dataset size, or the level of dataset refinement, making them inadequate for effectively supporting studies and the development of applications in classical Chinese poetry. To address these issues, we propose a method for constructing a large-scale and fine-grained multimodal knowledge graph of classical Chinese poetry. We first design an informative ontology graph for classical Chinese poetry and comprehensively collect knowledge about poetry based on it. Furthermore, the method utilizes knowledge augmentation, prompt optimization, and text-image alignment to acquire comprehensive and fine-grained knowledge. Both qualitative and quantitative evaluations are conducted on the Multimodal Knowledge Graph of Classical Chinese Poetry (CPMK), highlighting its comprehensiveness and high quality. We also conduct downstream evaluations on poetry-image retrieval, poetry question answering, and poetry theme classification tasks. Significant results were achieved in all three tasks, particularly in poetry-image retrieval and poetry theme classification, which attained state-of-the-art performance. This outstanding performance highlights the effectiveness of CPMK, which provides a robust foundation for research on classical Chinese poetry. CPMK will be released to promote research in Chinese culture[1].

## 1 Introduction

Classical Chinese poetry is a treasured cultural heritage that passes down ancient literature and fosters cross-cultural understanding between the East and the West. As times change, understanding Chinese poetry has become increasingly difficult. Differences between ancient poetry and modern Chinese, the evolution of poetry imagery meanings, and factors such as the poetry background all affect our understanding of Chinese poetry.

Numerous studies have shown that integrating information from multiple modalities can significantly improve the performance of downstream tasks. However, most existing research on classical Chinese poetry is predominantly focused on the textual modality(Wang et al., 2023b; Wei et al., 2024). The scarcity of multimodal datasets hinders research efforts that extend beyond the textual modality. Therefore, constructing a MultiModal Knowledge Graph (MM-KG) of classical Chinese poetry is essential for promoting Chinese culture and advancing research in this area.

To facilitate the discussion, we introduce several key concepts relevant to classical Chinese poetry. Ancient Poetry (AP) refers to classical Chinese poetry, characterized by its traditional form and archaic language, which differ significantly from modern Chinese. Poetry Imagery (PI) denotes specific objects or concepts that poets use to express emotions and thoughts. Imagery Meaning (IM) is the modern Chinese meaning of the PI, while Imagery Image (II) refers to the visual representation of the IM. Modern Chinese Translation (MCT) refers to the translation of classical Chinese poetry into modern Chinese. We provide an example in Appendix B to illustrate these concepts.

---

[1]https://github.com/***/CPMK

To the best of our knowledge, the currently available MM-KG of classical Chinese poetry is limited to PKG(Li et al., 2022), which is the only text-vision modality knowledge graph in this field. However, it has many shortcomings, making it difficult to support downstream tasks.

**1)** PKG focuses solely on PI-related knowledge, neglecting other critical aspects such as poetry-related knowledge and author-related knowledge, all of which are essential for poetry research. For instance, Jiang et al. (2024) utilizes poetry appreciation to assist in generating images of AP. **2)** IIs in the PKG are represented as URLs in website Unsplash[2], but many of these images are no longer available. Among 127,100 randomly sampled URLs, 9,837(7.7%) were found to be invalid, severely impacting the knowledge graph's utility in downstream tasks. **3)** For the acquired textual data, PKG does not perform sufficient data processing, so the data quality largely depends on the original websites and the crawler scripts, leading to a large amount of textual noise. Moreover, it fails to differentiate between the various IMs of the same PI. Using multiple IMs together as a query to retrieve IIs results in a weak correspondence between IMs and IIs. Currently, some studies (Liu et al., 2025) attempt to remove textual noise using regular expressions, but heuristic rules struggle to cover all cases in large-scale datasets. **4)** Chinese Poetry originated in the pre-Qin period (before 1000 BCE). Over time, the dissemination process may have led to variations in poetry-related knowledge. However, to the best of our knowledge, existing studies on classical Chinese poetry (Wei et al., 2024; He et al., 2023) overlooked these variations, leading to constructed datasets that lack completeness. **5)** Auditory elements are crucial components of classical Chinese poetry. These elements are mandated in many poetry forms, such as five-character and seven-character poems. However, the auditory data are overlooked in PKG.

To address the issues above, this paper proposes a novel method for constructing a large-scale and fine-grained MM-KG of classical Chinese poetry, which integrates textual, visual, and auditory modalities. To obtain comprehensive knowledge of classical Chinese poetry, we constructed an ontology graph encompassing multiple aspects of poetry-related knowledge. Guided by this graph, we systematically collected knowledge related to its concepts. To ensure the completeness of textual knowledge, we employ a poetry knowledge augmentation strategy. For the visual data in the ontology graph, we utilize generative models to acquire images, rather than traditional web scraping, enhancing the correlation between text and images. In the process of image generation, prompts are first processed by prompt optimization to improve the quality of generated images. For the obtained text-image pairs, text-image alignment is used to filter out high-quality text-image pairs. For auditory data, we gathered auditory knowledge for characters found in classical Chinese poetry. The proposed method leads to the construction of an MM-KG of Chinese Poetry (CPMK), which includes textual, visual, and auditory modalities.

Qualitative and quantitative evaluations demonstrate that CPMK is more comprehensive and accurate than existing datasets. To further validate the effectiveness of CPMK in downstream tasks, we incorporate it into poetry-image retrieval, poetry question answering, and poetry theme classification tasks. Experimental results demonstrate that CPMK significantly improves the performance of downstream tasks, particularly in poetry-image retrieval and poetry theme classification, where it attained state-of-the-art performance. Through qualitative research, quantitative research, and validation in downstream tasks, it demonstrates that CPMK can provide a solid foundation for the study and development of classical Chinese poetry applications. Our contributions are listed below:

- We propose a novel method for constructing a large-scale and fine-grained MM-KG of classical Chinese poetry. We first design an ontology of classical Chinese poetry to gather comprehensive knowledge, and adopt knowledge augmentation, prompt optimization, and text-image alignment to acquire a large-scale and fine-grained MM-KG.

- Using this method, we construct a multimodal knowledge graph of classical Chinese poetry with **6,834,825** textual nodes, **211,467** visual nodes, and **82,679** auditory nodes. Qualitative evaluation, quantitative evaluation, and downstream task validation collectively confirm its quality and effectiveness in the field of classical Chinese poetry.

- We construct two datasets for the classical Chinese poetry-image retrieval task using manual collection and automated generation methods. To our knowledge, this is the first benchmark for this task. They facilitate the evaluation of retrieval models and promote further research in the field of classical Chinese poetry.

---

[2]https://unsplash.com

- We propose a knowledge-enhanced poetry-image retrieval model. By establishing connections between classical Chinese poetry and images through modern Chinese translation of poetry, the model achieves state-of-the-art results on two datasets in the multimodal task of poetry-image retrieval. It uses a large amount of textual data and only a small amount (or even no) visual data, providing insights for other multimodal tasks.

- We validate the effectiveness of CPMK across multiple tasks by proposing a retrieval-augmented poetry question answering framework and a retrieval-augmented poetry theme classification framework. We achieve excellent results in both tasks, with the poetry theme classification achieving state-of-the-art performance.

## 2 RELATED WORKS

### 2.1 KNOWLEDGE GRAPH CONSTRUCTION

Due to advancements in LLMs, many studies have utilized them to construct knowledge graphs. Wang et al. (2025) leverages LLMs for triple extraction, relational embedding, and schema-based normalization, which supports multi-domain construction without retraining or fine-tuning. FolkScope(Yu et al., 2023) leverages the generative power of LLMs and human-in-the-loop annotation to semi-automatically construct the knowledge graph. However, in the field of classical Chinese poetry, the lack of a large-scale knowledge base like Wikipedia makes it difficult to collect substantial amounts of data, rendering existing methods difficult to apply directly.

In the field of classical Chinese poetry, there have also been studies focused on constructing knowledge graphs. KnowPoetry (Hong et al., 2020) proposes a framework to extract poems, poets, and their relationships from Tang poetry, thereby constructing a domain ontology and a knowledge graph. SKG-Poetry (Zhao et al., 2022) constructs a sememe knowledge graph of classical Chinese poetry, linking classical and modern Chinese vocabularies to enhance semantic understanding. These knowledge graphs are either constrained by their modalities or suffer from quality deficiencies, which makes it difficult for them to support downstream tasks effectively.

### 2.2 CLASSICAL CHINESE POETRY DATA

Research on classical Chinese poetry data mainly focuses on text, with limited exploration of vision and audio modalities. The ancient corpora of text include four main datasets: Poetry(Werneror), CCPM(Li et al., 2021), ACP-Corpus(Liu et al., 2025), Chinese-poetry-and-prose(VMIJUNV).

There is limited attention to vision and audio modalities in the study of classical Chinese poetry. In terms of vision modality, the PKG (Li et al., 2022) compiles knowledge related to PI, and (Liu et al., 2020) maps poems to specific categories and collects images corresponding to those categories. Regarding the audio modality, to our knowledge, no relevant knowledge graph has been identified.

## 3 METHOD FOR CONSTRUCTING CPMK

We analyse the data requirements from recent studies on classical Chinese poetry, such as Li et al. (2022); Jiang et al. (2024); Li et al. (2021), to construct an ontology graph. This graph serves as the guidance for the construction of the MM-KG of classical Chinese poetry. The ontology graph is in Appendix C. Guided by the ontology graph, this method overcomes the limitations of previous studies, which lacked comprehensive coverage of Chinese poetry knowledge. The construction method of MM-KG of Chinese poetry is divided into the following parts:

### 3.1 ACQUISITION OF RAW DATA.

Knowledge related to AP and author is crawled from the authoritative poetry website SouYun[3]. We extract words that appear more than 5 times and all the characters that have appeared in AP. These words and characters are used to crawl for their semantic meanings on the website HanDian[4]. For

---

[3]https://sou-yun.cn/
[4]https://www.zdic.net/

words, if their semantic meaning exists, they are categorized as PI, and their meaning serves as IM. For characters, in addition to their semantic meanings, we also crawl their auditory knowledge and visual knowledge in HanDian. Characters are visually represented in either GIF or SVG to demonstrate the stroke order of writing. Pinyin and Zhuyin are offered as audio to illustrate pronunciation.

When dealing with II data, manual collection of extensive II data is impractical, and web scraping poses significant challenges due to the unique characteristics of classical Chinese poetry. 1) There is a lack of comprehensive image databases for Chinese literature, as existing large-scale image websites primarily focus on modern elements and offer limited coverage of ancient Chinese literature. 2) Some IMs are relatively abstract, making it challenging to find images that basically convey their visual meaning when using web scraping.

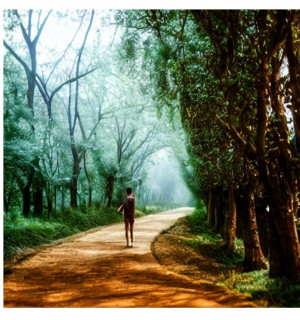 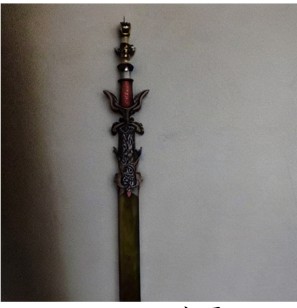

ChenMeng(尘梦)    ChiXiao(赤霄)

Figure 1: II for the "ChenMeng" and "ChiXiao", both generated using a generative model.

Generative models, having been trained on large-scale datasets, can effectively address the mentioned issues. For instance, the PI "ChiXiao(赤霄)"'s IM refers to the legendary ancient sword of "LiuBang(刘邦)", the PI "ChenMeng(尘梦)"'s IM symbolizes the illusion of the mortal world. As shown in Figure 1, generative models can generate content related to ancient legends and abstract concepts with relatively effective results. Therefore, this paper uses a generative model to create IIs. The selection of the generative model and the prompt setting can be found in Appendix D.

## 3.2 POETRY KNOWLEDGE AUGMENTATION

Currently, most data related to the AP comes from the internet, and its accuracy largely depends on the quality of the websites. Due to historical factors and the diverse transmission of poetry knowledge, variations and inconsistencies may exist. This paper adopts a cross-augmentation strategy, which integrates variations from multiple knowledge bases to provide the most comprehensive and reliable knowledge. We focus on two core aspects: knowledge related to AP and author.

We collect knowledge about AP and author from GuShiWen[5] and GuoXueHui[6]. For AP-related knowledge, we employ a two-phase deduplication strategy inspired by (Liu et al., 2025): global alignment removes redundant poems, while local alignment segments poems by punctuation and evaluates overlaps between text chunks. Similar APs are clustered rather than overwritten, with their relevant knowledge integrated to ensure a comprehensive representation. Details of the process are provided in Appendix E. For author-related knowledge, we determine entity consistency by verifying the author's name and dynasty, and then aggregate the relevant information.

## 3.3 PROMPT OPTIMIZATION FOR IMAGERY IMAGE GENERATION

Generative models often produce highly accurate images, but maintaining consistency with the input text remains a challenge. Many generative models utilize CLIP's text encoder as their text encoder, with its parameters frozen while only the diffusion process is trained (Ramesh et al., 2022; Rombach et al., 2022). However, research from Zhang et al. (2024) shows that CLIP's text encoder effectively handles fewer than 20 tokens, leading to hallucinations when processing longer texts.

---

[5]https://www.gushiwen.cn/
[6]https://www.gushicimingju.com/

Table 1: Average token distribution. SP Knowledge represents supplementary knowledge. Visual Desc represents visual description.

| Data | Average Token Length | Total Num |
|------|---------------------|-----------|
| Raw IM | 21.72 | 177,664 |
| Refined IM | 6.70 | 257,028 |
| SP Knowledge | 23.72 | 58,827 |
| Visual Desc | 15.30 | 135,720 |

Table 1 shows that raw IMs from the internet exceed token limits(21.72). Raw IMs often contain multiple meanings that have not been correctly separated. In addition, text data obtained through web scraping may introduce irrelevant noise and extraneous information, such as the sources of IMs, which can aid in understanding complex meanings but often hinder downstream tasks. To address this, this paper uses LLMs to filter noise, separate complex Raw IMs into distinct meanings, and retain useful auxiliary information as supplementary knowledge, producing refined IMs.

Table 1 shows that the refined IMs often become overly concise(6.70), failing to achieve the optimal token length. Intuitively, providing detailed descriptions within the model's comprehension range enhances the accuracy of the generated images. For example, prompts like "the sea god" are too concise, whereas "The majestic sea god stands above the waves" offers a clearer and more interpretable context for the generative model. Drawing inspiration from Retrieval Augmented Generation(RAG) technologies, we utilize LLMs to rewrite refined IMs into visual descriptions suited for generative models. This method enhances clarity and relevance while keeping the prompt length within a manageable 20 tokens, effectively reducing the likelihood of hallucinations.

LLMs are also used to determine if an IM can be visually represented, discarding inputs like stopwords that lack visual significance. This ensures that only visually meaningful data is processed by the generative models, enhancing efficiency. The instruction is shown in Appendix G.

### 3.4 Imagery Meaning-Imagery Image Alignment.

When handling large-scale text-image pairs, aligning them accurately becomes a significant challenge. It is common to use the CLIPScore(Hessel et al., 2021) to evaluate the relevance between text and images. CLIPScore has certain limitations: a high threshold may lead to the omission of entities, while a low threshold can weaken the alignment between text and images, particularly when dealing with large-scale text-image pairs. Inspired by GLIDE (Nichol et al., 2022), which evaluates image generation quality through classification, we abandon the traditional threshold-setting approach. Instead, we propose leveraging an image-to-text retrieval task to address this alignment challenge.

In the image-to-text retrieval task, text perturbations are introduced. Specifically, for each generated II, II is used to retrieve IM along with the two text perturbations. The first perturbation randomly selects another IM from the total set of IMs, while the second perturbation is composed of a random character selected from the tokenization vocabulary in BERT(Devlin et al., 2019). If all of the generated IIs correctly retrieve the candidate IM, the IIs and IM are considered aligned, and the text-image pair with the highest CLIP score is selected as the final match. Otherwise, those text-image pairs are deemed mismatched and discarded.

## 4 Qualitative and Quantitative Evaluations

### 4.1 Quantitative Evaluation

1) We counted the number of entities in each dataset, with the results presented in Table 2.. To our knowledge, CPMK is the first dataset to integrate text, vision, and audio modalities within classical Chinese poetry. According to the table, CPMK significantly exceeds prior research in the number of entities. Large-scale datasets serve as a robust foundation for advancing research and application development in classical Chinese poetry.

2) To evaluate the effectiveness of the proposed prompt optimization and text-image alignment methods, we design a comparative evaluation. The raw IMs are optimized using a heuristic approach as

Table 2: Modal entity statistics across datasets. Results with * are inferred from their papers due to dataset unavailability.

| Corpus | #Text | #Vision | #Audio |
|---|---|---|---|
| CCPM[19] | 136,090 | - | - |
| RPG*[13] | 215,227 | - | - |
| VMIJUNV[30] | 1,515,463 | - | - |
| ACP-Corpus*[22] | 2,159,920 | - | - |
| PKG[20] | 111,5143 | 96,049 | - |
| Image2Poem*[21] | 117,867 | 1,036 | - |
| CPMK | 6,834,825 | 211,467 | 82,679 |

prompts to generate new images, which are then refined through our text-image alignment method. The image generation part is the same as that used in this paper. We recorded the average token length of the text information processed by the heuristic methods and calculated the CLIPScore[7] of the generated images. The details of the heuristic approach are shown in Appendix I.

The results in Table 3 demonstrate that images processed by prompt optimization have a higher CLIPScore compared to the heuristic approach. By simply reducing IM-II pairs from 407,160 to 319,419, text-image alignment significantly enhanced CLIPScore, proving its effectiveness. Notably, the final counts of IM-II pairs obtained through the heuristic approach (102,090) and prompt optimization (106,473) are very close. This suggests that LLMs with visual capabilities can effectively determine whether an IM is visually representable, enhancing computational efficiency. It also demonstrates that LLMs possess the ability to rewrite text prompts for image generation.

Table 3: The average CLIPScore of IM-II pairs. Heu stands for Heuristic approach, PO stands for Prompt Optimization, Align stands for the proposed text-image alignment, and Hig stands for the highest CLIPScore text-image pair among the generated images.

| Data | CLIPScore | Total Num |
|---|---|---|
| Heu | 0.912 | 752,772 |
| Heu + Align | 1.056 | 306,270 |
| Heu + Align + Hig | 1.068 | 102,090 |
| PO | 1.022 | 407,160 |
| PO + Align | 1.136 | 319,419 |
| PO + Align + Hig | 1.191 | 106,473 |

## 4.2 QUALITATIVE EVALUATION

In the qualitative evaluation, we designed a questionnaire to evaluate two aspects: the relevance between IM and II, and whether IM is reasonably split on CPMK and PKG. The relevance score and coverage score are used, both of which are scored from 0 to 5.

**Relevance Score:** This metric evaluates the connection between the IM and II, factoring in the II's quality. Full points are given if the II captures any essential meaning of the IM, with deductions for discrepancies. If there are two meanings and one is perfectly captured, full points 5 are awarded.

**Coverage Score:** Ranging from 0 to the relevance score of the current image, this metric measures the image's coverage of IM. Full score indicates complete coverage, while partial coverage results in proportional deductions. If one of two meanings is perfectly captured, a score of 2.5 is given.

The relevance score minus the coverage score can help determine whether the segmentation of the IM is reasonable. We invited five university students knowledgeable about classical Chinese poetry to evaluate 500 IM-II pairs randomly selected from each dataset, totaling 1000 pairs.

As shown in Figure 2, CPMK significantly outperforms PKG in both relevance and coverage scores, with a smaller gap between the two compared to CPMK, indicating that CPMK achieves more

---

[7]In this paper, we calculate CLIPScore using the CN-CLIP-1B model (Yang et al., 2022).

reasonable image segmentation. Additionally, the CLIPScore between IM and II further validates the higher similarity in CPMK. We provide a questionnaire example in Appendix F.2.

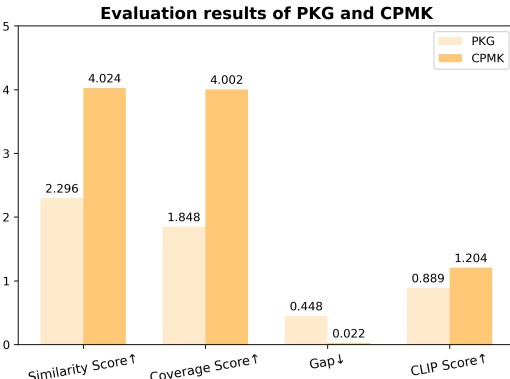

Figure 2: The evaluation results of IM-II for PKG and CPMK. Gap represents the Similarity Score minus the Coverage Score.

## 5 DOWNSTREAM TASK VALIDATION

To validate the effectiveness of CPMK in downstream tasks, we apply it to three downstream tasks: Poetry-Image Retrieval, Poetry Question Answering, and Poetry Theme Classification. Since our primary aim is to verify whether CPMK could play a role in downstream tasks, we don't focus on complex experimental designs. We conduct a preliminary experimental design and expect to achieve promising results to highlight the effectiveness of CPMK. Since both poetry question answering and poetry theme classification adopt retrieval-augmented techniques, without loss of generality, we introduce poetry question answering in the Appendix A.1.

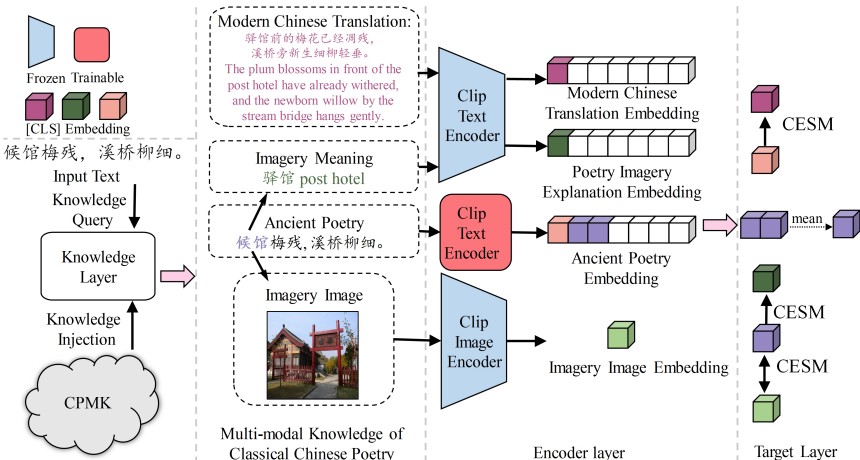

Figure 3: The overview framework of KPIR.

### 5.1 POETRY-IMAGE RETRIEVAL

While current retrieval models perform well in capturing the correspondence between text and images in modern languages, they face significant challenges in the poetry-image retrieval task. We attribute this primarily to the lack of domain-specific expertise in classical Chinese poetry. Furthermore, the limited size of existing Chinese poetry-image pair datasets poses a challenge for leveraging large-scale training methods such as CLIP (Radford et al., 2021). Based on the above analysis, we propose a Knowledge-enhanced Poetry-Image Retrieval model (KPIR).

Because the current retrieval model establishes a correspondence between modern Chinese and images, we leverage this by associating the encoding of the AP with its Modern Chinese Translation (MCT) to bridge the gap between APs and images. The core idea is to optimize the text encoder while keeping the image encoder fixed, ensuring that the embeddings of the AP align closely with its corresponding MCT. The structure of KPIR is shown in Figure 3.

Our model retrieves relevant knowledge from CPMK, including PI, IM, II, and MCT, forming the foundation for learning poetry expertise. We utilize a trainable text encoder and a frozen text encoder, along with a frozen image encoder. The frozen text encoder extracts MCT feature $f_{mct}$ and IM feature $f_{im}$, while the trainable text encoder extracts AP feature $f_{ap}$ and PI feature $f_{pi}$. The image encoder extracts the II feature $f_{ii}$. Knowledge injection is implemented through Cross-Entropy Similarity Matching (CESM), which integrates similarity scores of bimodal embeddings into a cross-entropy framework, minimizing the difference between the predicted and true distributions.

Given a mini-batch containing $N$ bimodal (X, Y) pairs, where Y includes poetry knowledge from CPMK (MCT, IM, II), based on either the AP or the PI in X. We form representation pairs $\{(f_i^x, f_j^y), y_{i,j}\}$ with labels $y_{i,j}$: 1 for matching pairs and 0 for non-matching ones. In a mini-batch, the CESM loss from modality X to Y is:

$$\text{CESM}(X,Y) = -\frac{1}{N}\sum_{i=1}^{N}\sum_{j=1}^{N} y_{i,j}\log\left(\frac{\exp(\text{sim}(f_i^x, f_j^y))}{\sum_{k=1}^{N}\exp(\text{sim}(f_i^x, f_k^y))}\right).$$

We applied the CESM in four stages: $\mathcal{L}_{ap2mct} = \text{CESM}(f_{ap}, f_{mct})$, $\mathcal{L}_{pi2im} = \text{CESM}(f_{pi}, f_{im})$, and $\mathcal{L}_{pi2ii} = \text{CESM}(f_{pi}, f_{ii})$ ,$\mathcal{L}_{ii2pi} = \text{CESM}(f_{ii}, f_{pi})$. Knowledge is injected through MCT($\mathcal{L}_{ap2mct}$) and IM($\mathcal{L}_{pi2im}$). We utilize $\mathcal{L}_{ii2pi}$ and $\mathcal{L}_{pi2ii}$ to preserve the correspondence between text and image. The final loss function is:

$$\mathcal{L} = \mathcal{L}_{ap2mct} + \mathcal{L}_{pi2im} + \mathcal{L}_{pi2ii} + \mathcal{L}_{ii2pi}$$

**Dataset and Evaluation Metrics.** We extract 30,000 pairs of AP and MCT from CPMK as the training dataset, with all PI-related knowledge sourced from CPMK. Due to the lack of existing datasets for poetry-image retrieval tasks, we construct two datasets for evaluation: 1) We manually collected 70 high-quality pairs of AP and image (PI-Manual) from the internet. 2) We generate 500 images corresponding to APs using a generative model (PI-Generate), with specific generation details provided in the appendix H. Since there is a one-to-one correspondence between text and image, this study uses recall as the evaluation metric, counting the number of correct answers within the retrieved set. The higher the recall, the better the model performs.

Table 4: Performance of Different Models on Poetry-Image Retrieval Tasks (†: MCT-Image Retrieval Tasks. w/o II: Training conducted without II, utilizing only text data.)

| Models | PI-Manual | | PI-Generate | | | | | | |
| | t2i | i2t | t2i | | | i2t | | | |
| | R@3 | R@3 | R@5 | R@10 | R@20 | R@5 | R@10 | R@20 |
|---|---|---|---|---|---|---|---|---|
| Taisu-0.2B[23] | 0.714 | 0.700 | 0.281 | 0.370 | 0.494 | 0.284 | 0.384 | 0.474 |
| AltClip-0.9B[7] | 0.601 | 0.671 | 0.288 | 0.368 | 0.484 | 0.314 | 0.412 | 0.492 |
| D2D2-0.4B[37] | 0.586 | 0.814 | 0.198 | 0.258 | 0.324 | 0.348 | 0.452 | 0.558 |
| CN-CLIP-0.1B[39] | 0.714 | 0.729 | 0.208 | 0.294 | 0.382 | 0.282 | 0.370 | 0.492 |
| CN-CLIP-0.4B[39] | 0.686 | 0.771 | 0.214 | 0.274 | 0.348 | 0.352 | 0.452 | 0.538 |
| CN-CLIP-1B[39] | 0.743 | 0.714 | 0.218 | 0.288 | 0.358 | 0.308 | 0.384 | 0.482 |
| CN-CLIP-0.1B[39]† | 0.871 | 0.889 | 0.372 | 0.468 | 0.576 | 0.390 | 0.476 | 0.574 |
| CN-CLIP-0.4B[39]† | **0.886** | 0.887 | 0.388 | 0.464 | 0.536 | 0.414 | 0.492 | 0.596 |
| CN-CLIP-1B[39]† | 0.871 | **0.901** | 0.376 | 0.446 | 0.534 | 0.408 | 0.498 | 0.588 |
| KPIR-0.1B w/o II | 0.857 | 0.728 | 0.422 | 0.514 | 0.642 | 0.294 | 0.384 | 0.498 |
| KPIR-0.1B | 0.871 | 0.800 | 0.460 | 0.548 | 0.644 | 0.340 | 0.448 | 0.554 |
| KPIR-0.4B | **0.886** | 0.871 | 0.500 | 0.594 | 0.682 | 0.458 | 0.548 | **0.664** |
| KPIR-1B | 0.857 | 0.871 | **0.508** | **0.602** | **0.700** | **0.462** | **0.560** | 0.656 |

**Main results.** As shown in Table 4, we evaluate two tasks: the core poetry-image retrieval task and the MCT-image retrieval task. The results of the latter are marked with a †. The MCT-image retrieval task uses modern Chinese translations of ancient poetry to perform bidirectional retrieval with images. The experimental results on poetry-image retrieval demonstrate that KPIR achieves state-of-the-art performance on the poetry-image retrieval task across two datasets. Even our smallest model, KPIR-0.1B, significantly surpasses previous methods. By integrating MCT knowledge and PI-related knowledge through CESM, KPIR effectively aligns the embeddings of APs with their corresponding MCTs, thereby establishing associations between APs and images. Notably, we also try training without II, utilizing only text data ($\mathcal{L} = \mathcal{L}_{ap2mct} + \mathcal{L}_{pi2im}$). The experimental results also outperform previous models, indicating that our proposed knowledge injection through poetry translation is effective.

KPIR is initialized using the CN-CLIP(Yang et al., 2022) model. On the PI-Manual dataset, KPIR's poetry-image retrieval performance is comparable to CN-CLIP's MCT-image retrieval. However, on the PI-Generate dataset, KPIR significantly outperforms CN-CLIP. This result is counterintuitive, as it would be reasonable to expect CN-CLIP to achieve superior performance, given that it was trained on a large-scale contemporary Chinese dataset. This indicates that if KPIR relied solely on MCT knowledge, its performance would be upper-bounded by that of CN-CLIP. KPIR's superior performance is attributed to both MCT knowledge and PI-related knowledge.

## 5.2 POETRY THEME CLASSIFICATION

Poetry Theme Classification is a fundamental task in classical Chinese poetry research, involving categorizing poems based on their themes. We applied CPMK and PKG to this task and evaluated their performance using the TCCP[8] dataset. TCCP is a theme classification dataset for Chinese classical poetry, which contains 3,247 poems. Its theme is divided into nine categories: homesickness, chanting things, landscape, missing someone, meditating on the history, pastoral, frontier war, boudoir resentment, and farewell. We use PI in the poetry as a query to retrieve relevant IM from CPMK and PKG, and then combine this knowledge with the original poem to input into DeepSeek-Chat(Guo et al., 2025) for poetry theme classification. Details are in Appendix A.5.

Table 5: The performance of different models on TCCP, with * indicating results are cited.

| Models | Micro-F1 | Macro-F1 |
|---|---|---|
| BERT+FT* [8] | 67.98 | 65.46 |
| HiAGM-TP*[48] | 63.02 | 57.18 |
| LCM*[10] | 86.39 | 85.11 |
| GreaseLM*[46] | 74.87 | 73.90 |
| KPT*[16] | 80.19 | 82.05 |
| ChatGLM*[43] | 88.04 | 85.15 |
| DeepSeek-Chat[11] | 89.12 | 87.23 |
| DeepSeek-Chat +PKG | 92.59 | 90.75 |
| DeepSeek-Chat +CPMK | **94.24** | **92.71** |

The results indicate that both CPMK and PKG enhance the model's classification capability. However, CPMK demonstrates a more significant improvement, achieving state-of-the-art results. This suggests that when inputting the same type of knowledge, CPMK provides both higher accuracy and a more comprehensive coverage than PKG.

## 6 CONCLUSION

This paper proposes a method for constructing an MM-KG for classical Chinese poetry, integrating textual, visual, and auditory modalities. By introducing knowledge augmentation, we ensure textual data completeness. We enhance the correlation between text and images through prompt optimization and text-image alignment. Qualitative evaluation, quantitative evaluation, and downstream tasks evaluation validate the quality and effectiveness of CPMK. The CPMK will be open-sourced to promote the development of the field of ancient poetry.

---

[8]https://github.com/shuizhonghaitong/classification_GAT/tree/master/data

## 7 REPRODUCIBILITY STATEMENT

To enhance the reproducibility of the knowledge graph construction and downstream tasks, we provide detailed descriptions of our methodologies. For II generation, Appendix D covers the selection of generation models and the setting of prompts. Regarding prompt optimization, the selection of LLM and prompt design can be found in Appendix G. Appendix 1 describes the algorithm for merging relevant knowledge for knowledge augmentation. In Section 3.4, we explain how text perturbations are constructed and the evaluation metrics used for text-image alignment. Finally, Appendix A includes the experimental designs for poetry-image retrieval, as well as the prompt settings for the poetry theme classification task and the poetry question-answering task.

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

# A  DETAILS OF DOWNSTREAM TASKS VALIDATION

## A.1  PERFORMANCE OF POETRY QUESTION ANSWERING TASK

To validate the comprehensiveness of the data in CPMK, we applied it to the poetry question answering. Following the traditional RAG framework, we use an LLM to extract the author, title, and AP as keywords from the original query. Like LightRAG(Guo et al., 2024), we use keywords as query conditions to retrieve related knowledge from CPMK. The retrieved knowledge is combined with the original query and fed into the LLM to answer questions.

We evaluate five tasks related to classical poetry in WenMind(Cao et al., 2024), including Basic Q&A (T1), Ancient Poetry Translation (T2), Sentiment Classification (T3), Ancient Poetry to English (T4), and Poet Introduction (T5), totaling 1,310 questions. T1 involves questions related to the basic knowledge of ancient poetry, such as answering the title and author based on the content. T2 is about translating ancient poetry into modern Chinese. T3 deals with sentiment classification of the poetry. T4 involves translating the poetry into English. T5 provides an introduction to the poet. We conducted experiments on ChatGPT-4, using the same model scoring metric from WenMind.

Table 6: The performance of different models, with * indicating results cited from original paper.

| Models | T1 | T2 | T3 | T4 | T5 |
|---|---|---|---|---|---|
| ChatGLM3-6B*[9] | 10.6 | 55.5 | 43.0 | 44.9 | 33.3 |
| Ancient-Chat-7B*[3] | 14.7 | 52.7 | 36.0 | 28.6 | 23.9 |
| LLaMA3-Chinese-8B*[1] | 1.7 | 62.4 | 42.5 | 52.3 | 23.4 |
| Baichuan2-13B-Chat*[38] | 20.1 | 66.9 | 43.0 | 51.3 | 55.4 |
| Ziya-LLaMA-13B*[2] | 6.4 | 57.5 | 40.5 | 40.2 | 31.6 |
| Qwen1.5-32B-Chat*[5] | _32.0_ | 67.9 | _64.0_ | 54.7 | 58.1 |
| Yi-1.5-34B-Chat*[41] | 30.5 | 69.0 | 53.5 | 54.2 | _64.6_ |
| ChatGPT-4[4] | 23.6 | **75.9** | 61.3 | _66.3_ | 44.1 |
| ChatGPT-4-RAG | **73.4** | _75.8_ | **64.3** | **71.2** | **65.9** |

As shown in Table 6, our ChatGPT-4-RAG demonstrates significant performance across most tasks, thanks to the high quality of CPMK, which enhances the model's understanding of ancient poetry. However, its performance on T2 is inferior to that of ChatGPT-4, likely due to the overlap between WenMind's internet-based dataset and the extensive datasets used for training current LLMs. We think the results are sufficient to illustrate the quality of CPMK. This study employed a simple RAG framework without task-specific adjustments, yet achieved significant performance improvements in most tasks. The experimental results demonstrate that CPMK is of high quality and can provide the model with better knowledge related to classical Chinese poetry.

## A.2  POETRY-IMAGE RETRIEVAL TASK IMPLEMENT DETAILS

KPIR is initialized using the CN-CLIP (Yang et al., 2022) model. We have experimented with different scales of the CN-CLIP model. We use the Adam optimizer (Kingma, 2014) with a weight decay rate of 0.01 and a learning rate of 2e-5. The seed is set to 123. Since our CESM model involves comparisons between mini-batches, we shuffle the order of the training set at the end of each epoch. Our experiments are conducted on an RTX 4090 24 GB GPU.

## A.3  POETRY-IMAGE RETRIEVAL TASK ABLATION STUDY

We conduct an ablation study to demonstrate the impact of each loss function on the poetry-image retrieval task. The experimental results are shown in Table 7. Without loss of generality, we conduct ablation experiments on the CN_CLIP_0.1B model. The results indicate that incorporating both global poetry translation knowledge and local PI-related knowledge leads to a significant improvement in the model's performance, particularly enhancing its ability to retrieve text from images while maintaining its poetry-image retrieval capability as much as possible. Even when trained solely on text, the model's performance also improves. We attribute this improvement to the preservation of text-image correspondence in CN_CLIP during fine-tuning, as well as our focus on fine-tuning the textual side, which correctly outputs poetry embeddings. Additionally, $\mathcal{L}_{pi2im}$ plays a crucial role

in the knowledge injection process. Given the abundant presence of PI in poetry, aligning PI with its meaning enables the model to accurately understand the poetry at a fine-grained level.

Table 7: Ablation experiments on the poetry-image retrieval task.

| loss function | | | | PI-Manual | | PI-Generate | | | | | |
| --- | --- | --- | --- | --- | --- | --- | --- | --- | --- | --- | --- |
| | | | | t2i | i2t | t2i | | | i2t | | |
| $\mathcal{L}_{pi2ii}$ | $\mathcal{L}_{ii2pi}$ | $\mathcal{L}_{ap2mct}$ | $\mathcal{L}_{pi2im}$ | R3 | R3 | R5 | R10 | R20 | R5 | R10 | R20 |
| | | ✓ | ✓ | 0.857 | 728 | 0.422 | 0.514 | 0.642 | 0.294 | 0.384 | 0.498 |
| | ✓ | ✓ | ✓ | 0.843 | 0.757 | 0.428 | 0.542 | **0.664** | 0.298 | 0.406 | 0.5169 |
| ✓ | | ✓ | ✓ | **0.914** | 0.757 | **0.476** | 0.550 | 0.646 | 0.294 | 0.041 | 0.512 |
| ✓ | ✓ | | ✓ | 0.886 | 0.771 | 0.468 | **0.558** | 0.656 | 0.280 | 0.418 | 0.514 |
| ✓ | ✓ | ✓ | | 0.643 | 0.714 | 0.264 | 0.376 | 0.474 | 0.276 | 0.356 | 0.438 |
| ✓ | ✓ | ✓ | ✓ | 0.871 | **0.800** | 0.460 | 0.548 | 0.644 | **0.340** | **0.488** | **0.544** |

## A.4 POETRY-IMAGE RETRIEVAL CASE STUDY

Figure 4 presents two case studies for the poetry-image retrieval task. In these cases, a significant discrepancy exists between the literal meaning and the poetry translation, posing a challenge to the model's retrieval abilities. KPIR overcomes this challenge by leveraging both PI-related and MCT knowledge to achieve a more accurate understanding of the ancient poetry. In the figure, areas associated with PI-related knowledge are highlighted with yellow solid lines, while those related to MCT-knowledge are marked with brown dashed lines.

For example, as shown in Figure 4 (a), PI-related knowledge such as "di (笛)" and "guan (管)" is often translated as flute or other musical instruments in Modern Chinese. However, in the context of ancient poetry, they are more literally associated with bamboo. PI like "xinhuang(新篁)" are rarely used in contemporary language. These linguistic factors increase the difficulty of comprehending ancient poetry. By learning PI-related knowledge, KPIR overcomes these challenges and correctly interprets the meaning. In Figure 4 (b), MCT knowledge provides a crucial semantic supplement for the PI-related knowledge. Although the water corresponding to "xihu(西湖)" and "chunshui(春水)" is blue, the MCT knowledge introduces the concept of "green water" through poetry translation, thereby deepening the model's understanding of the ancient poetry.

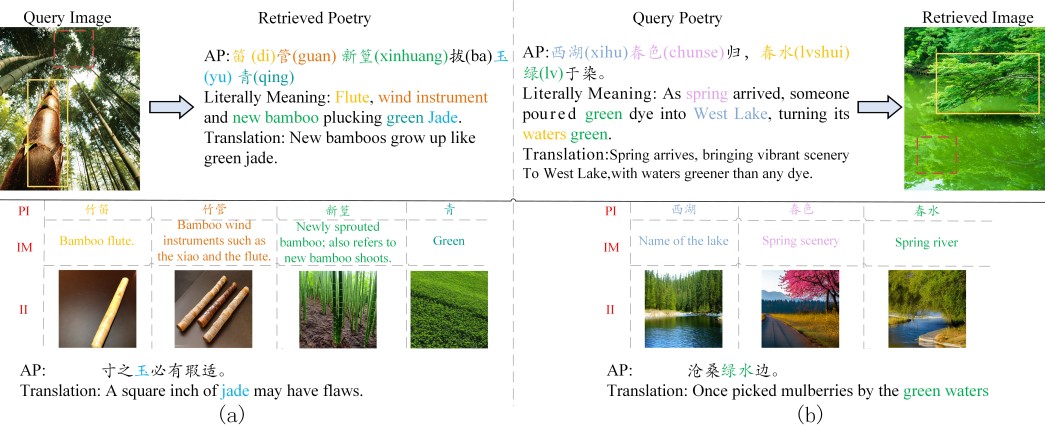

Figure 4: Two case studies of KPIR in the poetry-image retrieval task.

## A.5 POETRY THEME CLASSIFICATION TASK IMPLEMENT DETAILS

For the TCCP dataset, we divide it into the training, validation, and test sets in a ratio of 7:2:1. We use DeepSeek-Chat (Guo et al., 2025) as the base classification model. To enhance the model's ability to comprehend and analyze problems, we ask it not only to answer questions but also to provide explanations for its answers. Given that the TCCP dataset contains nine categories, we provide one learning example for each category, utilizing few-shot learning to improve the model's performance. The instruction to DeepSeek-Chat is

*You are an expert in Chinese classical poetry. The user will provide a series of questions related to themes of classical poems. Your answer categories should be within ["homesickness", "chanting things", "landscape", "missing someone", "meditating on the history", "pastoral", "frontier war", "boudoir resentment", and "farewell"]. The answer should only include the category and an explanation for your choice. For example, respond in the format: {"Answer": "boudoir resentment", "Explanation": "This poem expresses longing for a loved one far away, which is typical of the 'boudoir resentment' category."}*

To establish a standard evaluation, we use the same questions in TCCP as queries to retrieve relevant IM from CPMK and PKG, and use ChatGPT-4 to generate explanations for few-shot learning. This involves analyzing questions, integrating relevant knowledge from PKG or CPMK, and their answers to provide a coherent explanation. The instruction to ChatGPT-4 is

*As an expert in Classical Chinese poetry, you need to analyze the poetry theme classification problem provided by the user, along with the classification result. By considering the relevant poetry imagery and imagery meaning in the poem, explain why the poem fits the given theme and summarize the final result in a single paragraph.*

We provide a sample learning example for both CPMK and PKG in 5 and 6. System prompts are indicated with a brown dashed line, while learning samples are represented with a blue solid line.

**Role:System**
**Content:**你是一位中国古典诗歌专家，用户将提供一系列古诗主题分类问题，你的回答类别应该在['战争'，'送别'，'闺怨'，'咏物'，'怀古'，'田园'，'山水'，'思乡'，'怀人']内，答案应当只包含类别并解释为什么，例如以{"Answer":"闺怨"，"explanation"："这首诗表达了对远方亲人的思念之情，属于闺怨类诗歌。"}的形式返回。

**Role:User**
**Content:**
{"诗歌题目"："君山怀古"，"诗歌内容"："属车八十一，此地阻长风。千载威灵尽，赭山寒水中。"，
"诗歌意象&意象含义"：[{"属车"：["帝王出行时的侍从车"，"借指帝王"]}，{"十一"：["中华人民共和国的国庆节"，"十月一日"]}，{"长风"：["远风"，"暴风：大风"]}，{"千载"：["千年"]}，{"威灵"：["声威；威势"，"神灵"]}，{"赭山"：["伐尽山上的树木"，"伐尽树木后光秃的山岭"，"山名，因土石呈赭色"]}]}

**Role:Assistant**
**Content:**
{"Answer"："怀古"，
"explanation"："这首诗通过描绘君山的景象，引发对历史的感慨和追忆。'属车八十一，此地阻长风'中，'属车'指帝王出行时的侍从车，象征着过去的帝王威势；'长风'选择'远风'的含义，可能指阻隔的时间和空间距离，暗示往昔的遥远。'千载威灵尽'中的'千载'意为千年，强调时间的流逝；'威灵'指昔日君王的声威，这里表达了过去的帝王之威早已消逝。'赭山寒水中'中，'赭山'选择山名因土石呈赭色的含义，描绘君山的风貌，'寒水中'则可能指寒冷的水，整体表现出一种历史的冷峻和恍惚感。这些意象共同构成了怀古的氛围，展现了诗人对历史变迁的感叹和对往昔繁华的追忆。因此，这首诗属于怀古类诗歌。"}

Figure 5: Learning example for poetry theme classification using CPMK.

**Role:System**
**Content:**你是一位中国古典诗歌专家，用户将提供一系列古诗主题分类问题，你的回答类别应该在['战争'，'送别'，'闺怨'，'咏物'，'怀古'，'田园'，'山水'，'思乡'，'怀人']内，答案应当只包含类别并解释为什么，例如以{"Answer":"闺怨"，"explanation"："这首诗表达了对远方亲人的思念之情，属于闺怨类诗歌。"}的形式返回。

**Role:User**
**Content:**
{"诗歌题目"："君山怀古"，"诗歌内容"："属车八十一，此地阻长风。千载威灵尽，赭山寒水中。"，
"诗歌意象&意象含义"："[{"属车"：["1.帝王出行时的侍从车。秦汉以来，皇帝大驾属车八十一乘，法驾属车三十六乘，分左中右三列行进。2.借指帝王。 随从的车辆。"]}，{"十一"：["十月一日是中华人民共和国的国庆节；新中国成立于一九四九年的这一天"]}，{"长风"：["1.远风。2.暴风：大风。 大风。"]}，{"千载"：["千年。形容岁月长久。 比喻年代久远。"]}，{"威灵"：["声威；威势:神灵 声威、声势。《文选·陈琳·为袁绍檄豫州》：「幕府奉汉威灵，折冲宇宙。」《文选·何晏·景福殿赋》：「不壮不丽，不足以一民而重威灵；不饰不美，不足以训后而来厥成。」也作「威棱」。"]}，{"赭山"："[1.谓伐尽山上的树木。2.指伐尽树木后光秃的山岭。3.山名。因土石呈赭色，故称。在浙江萧山东北。古与龛山隔钱塘江对峙，东接鳖子门，为江海门户，后江流北移，此山遂与龛山同处江南。4.山名。因土石呈赭色，故称。在安徽省芜湖市北郊。上有宋英宗治平二年建造的塔和相传为宋黄庭坚读书处的滴翠轩等名胜古迹，今已辟为公园。"]}]"}

**Role:Assistant**
**Content:**
{"Answer"："怀古"，
"Explanation"："这段诗通过"属车八十一"引入帝王出行的盛大场景，暗示昔日的辉煌与尊贵；"此地阻长风"意象描绘地理环境的险恶，进一步衬托出历史变迁之无情；"千载威灵尽"则明确表达时间的流逝和昔日权势的消逝；"赭山寒水中"通过对山景的描绘，象征着往昔的繁荣已被遗忘在自然的寒冷与孤寂中。整体意象通过对比昔日盛况和现状的冷清，深刻表现怀古的主题，即对历史辉煌的追忆与对时间无情流逝的感叹。因此属于怀古诗。"}

Figure 6: Learning example for poetry theme classification using PKG.

## B  POETRY CONCEPT ILLUSTRATION

We present an example of using multimodal information to understand poetry in Figure 7. "SuShi(苏轼)"'s poem "(水光潋滟晴方好,山色空蒙雨亦奇)" describes the beauty of nature in diverse weather conditions. The evolving meanings of poetry imagery(PI), such as 'KongMeng (空蒙)', alongside the background of ancient poetry, have made it difficult to fully understand the poem. By presenting Imagery Meaning (IM) through text and images, as well as introducing relevant information about the poetry background, people from diverse cultural backgrounds can gain a deeper understanding of classical Chinese poetry.

| AP: | 水光 潋滟 晴方好， 山色 空蒙 雨亦奇。 | | |
|---|---|---|---|
| MCT: | Under the sunlight, West Lake glistens with shimmering waves, appearing stunningly beautiful. On rainy days, the misty mountains around the lake fade in and out of view, creating a mysterious charm. | | |
| PI: | 水光 | 潋滟 | 空蒙 |
| IM: | 水面映现出的光色 **The reflected light and colors on the water's surface.** | 形容水波荡漾 **Describe the rippling of the water.** | 细雨迷茫的样子 **The hazy appearance of a light drizzle.** |
| II: | | | |
| APBG: | As HangZhou's Vice Magistrate (1071–1074), Su Shi wrote many poems about West Lake, including this in early 1073. | | |

Figure 7: An example illustrating the use of multimodal information to enhance the understanding of ancient poetry. AP (Ancient Poetry), MCT (Modern Chinese Translation), PI (Poetry Imagery), IM (Imagery Meaning), II (Imagery Image), and APBG (Ancient Poetry Background) are introduced.

## C  ONTOLOGY GRAPH

We present the ontology graph in Figure 8, which classifies concepts into key concepts and attributes based on their significance.

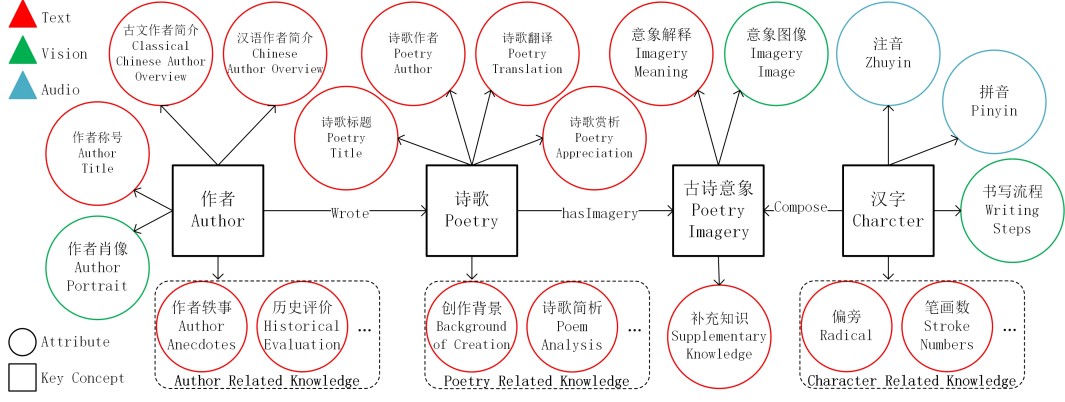

Figure 8: The ontology graph of classical Chinese poetry, where shapes represent types and colors represent modalities.

## D  GENERATIVE MODEL SELECTION

In this section, we discuss the selection of the image generation model and prompt settings. We chose the Taiyi-1B model (Zhang et al., 2022a) for its balance of quality and efficiency. As shown in Table 8, Taiyi-1B performs comparably to larger models while offering faster inference due to its smaller parameter size. However, due to computational limitations, the entire image generation process still took about a month.

Table 8: Comparison of different models in Chinese(COCO-CN) datasets. Data is cited from Wu et al. (2024)

| Models | CLIP Sim(↑) | FID(↓) | IS(↑) |
|---|---|---|---|
| Taiyi-1B[45] | 0.197 | 69.226 | 21.060 |
| Alt-1.5B[40] | 0.220 | 68.488 | 22.126 |
| Pai-1B[31] | 0.196 | 72.572 | 19.145 |
| Taiyi-3.5B[36] | 0.225 | 67.675 | 22.965 |

The text prompts were primarily set based on the model's recommendations. To generate realistic images reflecting common real-world scenes, we included the word "realistic (现实)" in positive prompts. However, as the model often generated anime-style images, we added "anime(动漫)" to steer it toward the desired output. The final positive prompt words are "({}, 现实)" where the IM is inserted into the {} position, while the negative prompt words are "(广告, ，, ！, ，。, ；, 资讯, 新闻, 水印, 动漫)".

## E  KNOWLEDGE AUGMENTATION

In this section, we detail the process of merging relevant data. First, APs are mapped to hash values to identify duplicates by comparing these hashes. Next, APs are segmented by punctuation marks ( ，。！？ ), and similar APs are merged based on the number of matching text segments. Duplicate or similar APs are merged, and their associated knowledge is integrated. This process is carried out according to Algorithm 1.

---

**Algorithm 1** Similarity Matching for the AP

---

**Input:** Raw databases $\{D_i\}$ , num of databases $l$.
**Output:** Deduplicated databases $D'$.
$P \leftarrow \bigcup_{i=1}^{l}$ Extract APs from $D_i$.
$D' \leftarrow \bigcup_{i=1}^{l}$ data from $D_i$.
**for** each pair $(P_s, P_l) \in P \times P$ **where** $P_s \neq P_l$ **and** $|P_l| \geq |P_s|$ **do**
   $N_s \leftarrow$ number of sentence segments in $P_s$
   $N_l \leftarrow$ number of sentence segments in $P_l$
   $C \leftarrow$ number of identical sentence blocks between $P_s$ and $P_l$
   **if** $C > \frac{N_s}{2}$ **then**
      $RK_s = \text{GetRelevantKnowledge}(P_s)$
      $D'[l] \leftarrow D'[l] \cup \{RK_s\}$
      $\text{Delete}(D'[s])$
   **end if**
**end for**

---

## F  QUESTIONNAIRE DETAILS

### F.1  QUESTIONNAIRE GUIDANCE

This section outlines the instructions provided to participants for evaluating the quality of knowledge graphs in our questionnaire. We provide guidelines for assessing two types of scores: similarity score and coverage score. As illustrated in Figure 9, we instruct participants to assign the highest similarity score (ranging from 0 to 5) if an image fully matches either the surface meaning (physical

description) or the deep meaning (emotional significance) of any given meaning. Partial matches require point deductions based on the degree of alignment. Figure 9 explains the scoring for coverage, where the similarity score is constrained to fall within the range of 0 to the relevance score assigned to the image. If the image only covers a subset of interpretations, deductions are made proportionally depending on the number and significance of the uncovered interpretations. To ensure clarity, we provide three illustrative examples for each type of score, addressing common scenarios.

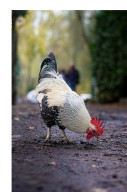

词语:鸡 **解释**:家禽
**Word**: Chicken **Definition**: Poultry

**Figure 1**

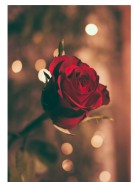

词语:爱 **解释**:1.对人或事有深挚的感情 2.容易
**Word**: Love **Definition**:
1.A deep affection for someone or something.
2.Easy to be prone to or inclined toward something

**Figure 2**

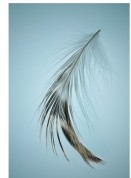

词语:驾鹊 **解释**:是传说牛郎、织女分居天河两岸，每年七夕，喜鹊飞临天河，……事见《岁华纪丽．七夕》注引汉应劭《风俗通》。后因以"驾鹊"为七夕的典实。
**Word**: Ride the Magpie **Definition**: Refers to the legend of the Cowherd and Weaver Girl….reuniting on Qixi Festival via a magpie bridge.

**Figure 3**

**相似性得分**:
如果图片符合词语的某一个解释的**表层含义-物理描述**或**深层含义-情感**，则相似性得分打满分。分值为0~5
如图1符合表层含义打5分，如图2玫瑰符合解释1的深层含义，虽不满足解释2"容易"，但较为满足解释1，打4分.
图3既不符合七夕的表层含义与深层含义打0分，若勉强符合可以凭感觉折中打分
**Similarity Score**:
If an image matches either the **surface meaning (physical description)** or **deeper meaning (emotional aspect)** of a word, it gets a full similarity score (0–5).
Examples:
•Image 1 matches the surface meaning —score: 5.
•Image 2 (e.g., a rose) fits the deeper meaning of Definition 1 but not Definition 2 ("easiness") —score: 4.
•Image 3 doesn't match Qixi's surface or deeper meanings —score: 0.
•If barely relevant, score based on intuition.

**覆盖性得分**:
**覆盖性得分的打分范围为(0，当前图像的相似性得分)**
如果图片能够覆盖该词语的所有解释上则赋予当前图像的相似性得分，否则根据覆盖度酌情减分
如图1符合所有解释打5分 ,图2因相关性得分为4，**得分范围为(0,4),因为只满足一个减半最终打2分**,
图3因相似性得分为0分则覆盖性也为0分。
**Coverage Score**:
The coverage score **ranges from 0 to the current similarity score of the image**. If the image covers all interpretations of the word, it gets the full similarity score. Otherwise, points are deducted based on the degree of coverage.
Examples:
•Image 1 fully covers all interpretations —score: 5.
•Image 2, with a similarity score of 4, has a coverage range of (0, 4). Since it only satisfies one interpretation, the score is halved to 2.
•Image 3, with a similarity score of 0, also gets a coverage score of 0.

Figure 9: Instructions for evaluating the Similarity Score and Coverage Score.

## F.2 QUESTIONNAIRE EXAMPLE

We provided a questionnaire example in Figure 10, illustrating the PI "QuShu(氍毹)" related knowledge in CPMK and PKG. In PKG, different meanings are distinguished by different colors, and grey indicates areas representing the origin of IM, which are treated as textual noise. Due to the influence of textual noise and improper segmentation of IM, the text-image correspondence in PKG is weak. The image barely reflects carpet, resulting in a relevance score of 2. Therefore, the coverage score for this text-image pair ranges from 0 to the relevance score (2), but it does not effectively convey the intended meaning of the stage, resulting in a compromised coverage score of 1. In CPMK, the correlation between IM and II is high, with both relevance and coverage scores of 4.5 for IM-II pairs.

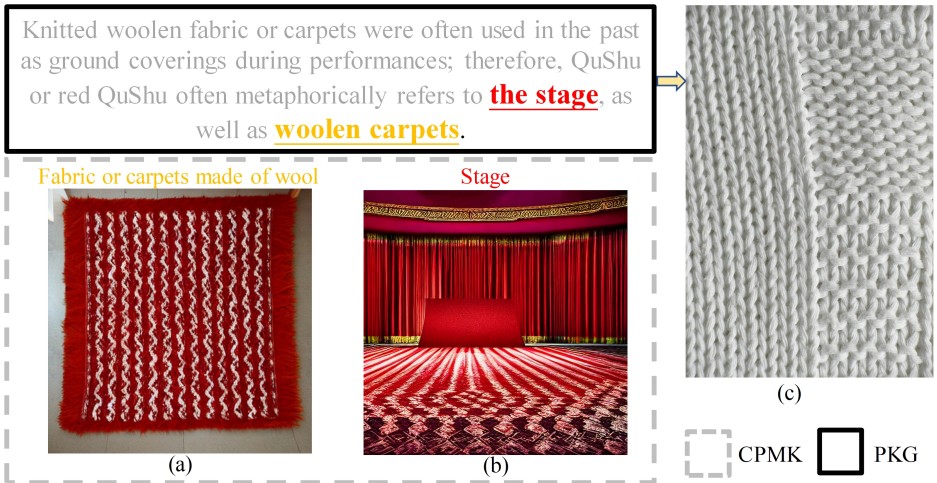

Figure 10: An example PI "QuShu(氍毹)" in the questionnaire.

# G  PROMPT OPTIMIZATION FOR IM

We introduce the instructions about prompt optimization for II generation, which is shown in Figure 11. We use DeepSeek-ChatGuo et al. (2025) for processing raw IMs.

 Input consists of keywords from classical Chinese poetry and their explanations. Each explanation may include multiple meanings as well as information about the poem's source.
**Input format**: Keyword&&Explanation
**Task**:
1. Data Cleaning:
    - Identify and extract the main meaning from the explanation.
    - Extract the poem's source, background explanation, and other related information as " Supplementary Knowledge". This information should retain its original format.
2. Determine if each main meaning has visual characteristics.
    - Visual characteristics refer to elements that can be visualized, such as natural landscapes, animals, plants, specific scenes, etc.
3. If a main meaning has visual characteristics, set " Visual Information " to true and generate a visual description no longer than 20 words.
    - The visual description should include details such as shape, color, action, background, etc., suitable for image generation.
 4. If a main meaning does not have visual characteristics, set "visual information" to false and leave "visual description" empty.
 **Return format should be as follows:**
 **Keyword**:[
{"Meaning": "Extracted main meaning 1", "Supplementary Knowledge": "Related background or source 1",
"Visual Information": true, "Visual Description": "Detailed visual description"},
{"Meaning": "Extracted main meaning 2", "Supplementary Knowledge": "Related background or source 2",
"Visual Information": false, "Visual Description": ""},]
**Example**:
Input: 剥床&&1.语出《易剥》:"剥床以足，以灭下也。"陈梦雷浅述:"侵灭正道，自下而上也。"又:"剥床以肤，切近灾也。"陈梦雷浅述:"阴祸已迫其身也。"后用"剥床"称残害忠良或迫身之祸。
**Output**: [
  {"Meaning": "残害忠良或迫身之祸", "Supplementary Knowledge": "语出《易剥》:"剥床以足，以灭下也。"陈梦雷浅述:"侵灭正道，自下而上也。"又:"剥床以肤，切近灾也。"陈梦雷浅述:"阴祸已迫其身也。",
"Visual Information": false, "Visual Description": ""}]

Figure 11: Instructions for processing raw IMs.

# H PI-GENERATE DATASET CONSTRUCTION

We detail the process of utilizing large language models (LLMs) to generate a poetry-image retrieval dataset(PI-Generate). Given the critical role of emotion in AP, we distinguish between scene and emotion descriptions to ensure the generated images accurately reflect the intended meanings of the poems. Specifically, this study retrieves modern Chinese translations and poetry appreciations of AP from CPMK. These texts are then refined using ChatGPT-4 Achiam et al. (2023), which generates tailored prompts that encapsulate both scene and emotion descriptions for image generation. Finally, DALL.E 3 Shi et al. (2020) leverages these tailored prompts to produce corresponding images. The detailed instructions for extracting scene and emotion descriptions are illustrated in Figure 12.

As an expert in Chinese classical poetry, please help me extract scene and emotion description from the translation and appreciation of Chinese classical poetry to use as prompts for image generation.
Input: {"translation":"Translation of the poetry",
 "appreciation":"Appreciation of the poetry"}
Output: {"scene_description":"Scene description from the poetry",
 "emotion_ description ":"Emotion description from the poetry"}
Requirements:
1. The scene and emotion description should be concise and clear.
2. The scene and emotion should be suitable as prompts for image generation models.
Example:
 Input: {"translation":"自古以来，人终不免一死！倘若能为国尽忠，死后仍可光照千秋，青史留名。",
 "appreciation":"此句悲壮激昂、掷地有声，以磅礴的气势、高亢的语调显示了诗人的民族气节和舍生取义的生死观，表达了诗人赤诚的爱国情怀和视死如归的崇高精神，激励了无数的爱国之士为了民族大业而抛头颅、洒热血。"}
 Output: {
 "scene_ description ":"壮烈的战场，战士奋勇牺牲，历史长河中的英雄光辉."
 "emotion_ description " :"悲壮激昂的爱国激情，视死如归的无畏精神。"}

Figure 12: Instructions for generating emotion description and scene description.

# I HEURISTIC APPROACH

This section introduces the heuristic approaches employed for processing raw IMs. The primary objective is to separate different IMs associated with the same PI, eliminate textual noise in PIs, and retain useful information from PIs as supplementary knowledge. Due to raw IMs with multiple meanings having various formats, it is challenging to develop heuristic rules that can be universally applied for splitting meanings. As a result, we split IMs based on the structure of the crawled data. Regarding textual noise, the data we gathered contains a large amount of noise. We use regular expressions to remove as much as possible. Regarding supplementary knowledge, we find that there is a significant amount marked by ( 《》 )(book title). However, due to the large volume of similar information, it is difficult to fully represent them using regular expressions alone. Here are three regular expression examples:

1) 语本《.*?》 ?: ".*?"

2) 王逸注:.*?。

3) 《.*?》： .*? "

## J EXAMPLES OF CPMK

In this section, we provide examples of the CPMK.

### J.1 EXAMPLE OF PI AND CHARACTER RELATED KNOWLEDGE IN CPMK

In Figure 13, we present an example of PI and character-related Knowledge in CPMK. For PI (谢墅), we provide its IM, Supplementary Information, and II. For the character (墅), we provide its Pinyin, ZhuYin, Stroke Count, and Explanation.

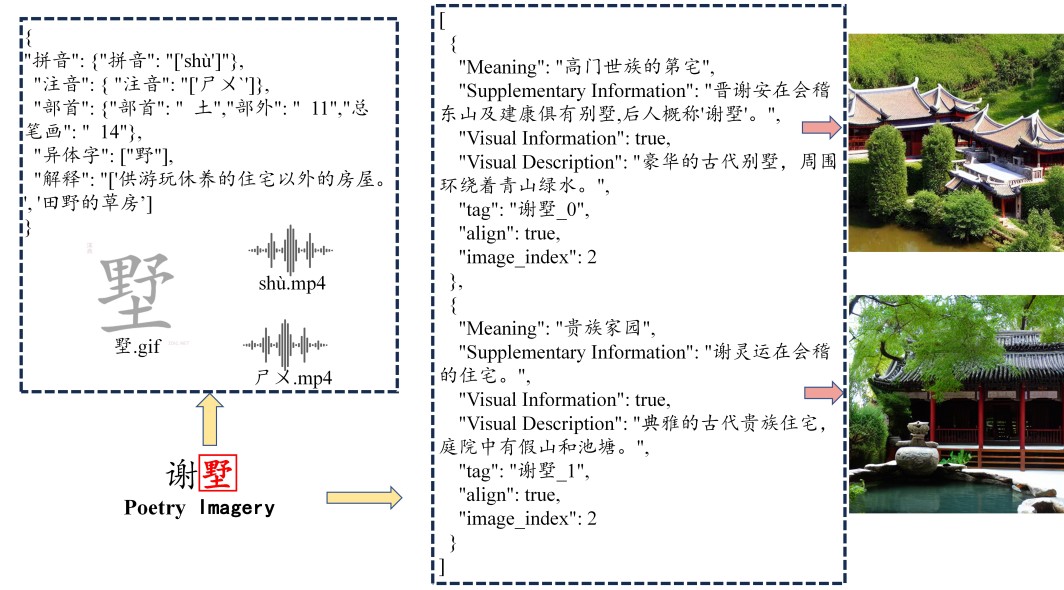

Figure 13: An example of character-related knowledge and PI-related knowledge in CPMK.

### J.2 EXAMPLE OF POETRY-RELATED KNOWLEDGE IN CPMK

In Figure 14, we present an example of poetry-related knowledge in CPMK. We integrate knowledge from SouYun, GuShiWen, and GuoXueHui, striving to present as complete a representation of poetry-related knowledge as possible through the consolidation of various databases. Notably, the version of the classical Chinese poem obtained from SouYun (两岸猿声啼不尽) differs from those found in GuShiWen and GuoXueHui (两岸猿声啼不住). Through the use of knowledge augmentation, we provide a comprehensive representation of the knowledge related to these poems.

### J.3 EXAMPLE OF AUTHOR-RELATED KNOWLEDGE IN CPMK

In Figure 15, we present an example of author-related knowledge in CPMK. This includes the poet's name(李白), the historical era, a brief introduction, and an image of the poet. Besides knowledge from SouYun, we also integrate various data sources from SouYun to provide a comprehensive introduction to the authors.

## K THE USE OF LARGE LANGUAGE MODELS

This paper utilizes LLMs as auxiliary writing tools. We used ChatGPT-4 to perform simple refinements on this paper, correcting basic grammatical and spelling errors.

[ { "**标题**": "早发白帝城（一作白帝下江陵）",
  "**作者**": "李白",
  "**朝代**": "唐朝",
  "**内容**": "朝辞白帝彩云间，千里江陵一日还。两岸猿声啼不尽，轻舟已过万重山。",
  "**来源**": "搜韵"},
 {"**标题**": "早发白帝城",
  "**内容**": "朝辞白帝彩云间，千里江陵一日还。两岸猿声啼不住，轻舟已过万重山。",
  "**朝代**": "唐朝",
  "**作者**": "李白",
  "**古诗简介**": "《早发白帝城》是唐代伟大诗人李白在流放途中遇赦返回时所创作的一首七言绝句，是李白诗作中流传最广的名篇之一。诗人是把遇赦后愉快的心情和江山的壮丽多姿、顺水行舟的流畅轻快融为一体来表达的。全诗无不夸张和奇想，写得流丽飘逸，惊世骇俗，但又不假雕琢，随心所欲，自然天成。",
  "**翻译/译文**": "清晨我告别高入云霄的白帝城；江陵远在千里船行只一日时间。两岸猿声还在耳边不停地啼叫；不知不觉轻舟已穿过万重青山。",
  "**注释**": "发：启程。白帝城：故址在今重庆市奉节县白帝山上。朝：早晨。辞：告别。彩云间：因白帝城在白帝山上，地势高耸，从山下江中仰望，仿佛耸入云间。白帝：今四川省奉节。江陵：今湖北荆州市。一日还：一天就可以到达；还：归；返回。猿：猿猴。啼：鸣、叫。住：停息。万重山：层层叠叠的山，形容有许多。",
  "**赏析**": "《早发白帝城》是唐代伟大诗人李白在流放途中遇赦返回时所创作的一首七言绝句，是李白诗作中流传最广的名篇之一。诗人是把遇赦后愉快的心情和江山的壮丽多姿、顺水行舟的流畅轻快融为一体来表达的。全诗无不夸张和奇想，写得流丽飘逸，惊世骇俗，但又不假雕琢，随心所欲，自然天成。……",
  "**古诗分类**": "[ '唐诗三百首', '小学生必背古诗70首', '七言绝句', '长江', '叙事诗', '山水诗']",
  "**来源**": "国学荟"},
 {"**标题**": "早发白帝城",
  "**朝代**": "唐朝",
  "**作者**": "李白",
  "**内容**": "朝辞白帝彩云间，千里江陵一日还。两岸猿声啼不住，轻舟已过万重山。",
  "**相关信息**": {
   "**译文及注释**": [
    {"**译文**": "清晨，我告别高入云霄的白帝城，江陵远在千里之外，船行只需要一天时间便能返回。两岸猿声还在耳边不停地回荡，轻快的小舟已驶过万重青山。" },
    {"**译文二**": "清晨，我告别高入云霄的白帝城，江陵远在千里，船行只需一日。两岸猿声，还在耳边不停地啼叫，不知不觉，轻舟已穿过万重青山。" },
    {"**注释**": "发：启程。白帝城：故址在今重庆市奉节县白帝山上。杨齐贤注：“白帝城，公孙述所筑。初，公孙述至鱼复，有白龙出井中，自以承汉土运，故称白帝，改鱼复为白帝城。”王琦注：“白帝城，在夔州奉节县，与巫山相近。所谓彩云，正指巫山之云也。”朝：早晨。辞：告别。彩云间：因白帝城在白帝山上，地势高耸，从山下江中仰望……"}
   ],
   "**创作背景**": "公元759年（唐肃宗乾元二年）春天，李白因永王李璘案被流放夜郎，途经重庆。行至白帝城的时候，忽然收到赦免的消息，惊喜交加，随即乘舟东下江陵。此诗即是作者回到江陵时所作，所以诗题一作《下江陵》。",
   "**赏析**": [
    {"**赏析**": "唐代安史之乱初期，唐玄宗奔蜀，太子李亨留讨安禄山，不久，李亨既位，史唐肃宗。玄宗又曾命令儿子永王李璘督兵平叛，永王李璘在江陵，召兵万人，自树一帜，肃宗怀疑他争夺帝位，已重兵相压，李璘兵败被杀。李白曾经参加过永王李璘的幕府，被加上“附逆”罪流放夜郎（今贵州遵义），……"}],
   "**简析**": "《早发白帝城》是一首七言绝句。此诗意在描摹自白帝至江陵一段长江水急流速、舟行若飞的情况。首句写白帝城之高；次句则描述了江陵路遥，舟行迅速；三句以山影猿声为背景，衬托行舟飞进；四句写行舟轻如无物，点明水势如泻。诗人遇赦后愉快的心情和顺水行舟的流畅轻快、……"},
   "**来源**": "古诗文"},
]

Figure 14: An example of poetry-related knowledge in CPMK

[ { "**诗人**": "李白",
  "**朝代**": "唐代",
  "**称号**": ["饮中八仙", "大李杜"],],
  "**介绍**": "李白（701年－762年），字太白，号青莲居士，又号"谪仙人"，祖籍陇西成纪（今甘肃省秦安县），出生于蜀郡绵州昌隆县（一说出生于西域碎叶）。唐代伟大的浪漫主义诗人，被后人誉为"诗仙"，与杜甫并称为"李杜"，为了与另两位诗人李商隐与杜牧即"小李杜"区别，杜甫与李白又合称"大李杜"。据《新唐书》记载，李白为兴圣皇帝（凉武昭王李暠）九世孙，与李唐诸王同宗。其人爽朗大方，爱饮酒作诗，喜交友。李白深受黄老列庄思想影响，有《李太白集》传世，诗作中多为醉时写就，代表作有《望庐山瀑布》《行路难》《蜀道难》《将进酒》《早发白帝城》等。",
  "**其他知识**": {
    "**轶事典故**": [ {"**友挚情**": "……" }, {"**生死考证**": "……"}],
    "**家庭成员**": [{ "**家人**": "……" },{ "**配偶**": "……"}, {"**子女**": "……"}],
    "**后世纪念**": [ {"**墓地**": "……" }, {"**纪念馆**": "……"}],
    "**主要成就**": [ {"**主要成就**": "……"}, {"**歌**": "……"}, { "**代表作品**": "……"}, {"**剑术**": "……"}, {"**道经**": "……" }, {"**思想**": "……"} ],
    "**人物生平**": [ {"**早年天才**": "……"}, {"**辞亲远游**": "……" }, {"**蹉跎岁月**": "……"}, {"**西游献赋**": "……"}, {"**李杜相识**": "……"}, {"**安史入幕**": "……"}, {"**溘然病逝**": "……"}],
    "**来源**": "古诗文"
  },
{"**其他介绍**":[
  {"**中国历代人名大辞典**": "【生卒】：701—762\n【介绍】：\n唐陇西成纪人，其先人隋末流寓西域，故生于安西都护府所属碎叶城。中宗神龙初，迁居蜀之绵州昌隆县青莲乡，又尝寓居山东，故亦称山……" },
  {"**唐诗大辞典修订本**": "【生卒】：701—762\n字太白，号青莲居士，排行十二，陇西成纪(今甘肃秦安西北)人，其先隋末窜于碎叶(今吉尔吉斯斯坦托克马克附近)，李白即出生于此。中宗神龙元年(705)随家……"},
  {"**唐诗汇评**": "李白（701-762），字太白，号青莲居士。祖籍陇西成纪（今甘肃秦安）。出生地有蜀中、西域、长安诸说，迄无定论。少时居绵州彰明县清廉乡（今属四川江油），读书吟诗，遍观百家……"},
  {"**词学图录**": "李白（701-762）字太白，号青莲居士。祖籍陇西成纪（今甘肃秦安东），隋末其先人流寓西域，白出生于安西大都护府碎叶城，五岁随父迁居绵州昌隆（今江油）青莲乡。天宝初供奉翰林。……"},
  {"**黄鹤楼志·人物篇**": "李白（701—762）唐代诗人。字太白，号青莲居士，世人又称谪仙、诗仙。祖籍陇西成纪（今甘肃静宁西南），先世流迁中亚，5岁随父定居绵州昌隆县（今四川江油县）青莲乡。……"},
  {"**全唐文·卷三百四十七**": "白字太白。兴圣皇帝九世孙。白生梦长庚星。因以命之。举有道不应。天宝初至长安。贺知章言于元宗。召见金銮殿。论当世事。奏颂一篇。诏供奉翰林。忤高力士。摘其诗激杨……"}
]}

Figure 15: An example of author-related knowledge in CPMK

