# OpenReview forum: "Foundation for Chinese Poetry Research: An Open Large-Scale and Fine-Grained Multimodal Knowledge Graph"
_ICLR.cc/2026/Conference — ICLR 2026 Conference Withdrawn Submission_

### Official Review · Reviewer_D9xt · 2025-10-22

**Soundness:** 2
**Presentation:** 3
**Contribution:** 2
**Rating:** 4
**Confidence:** 3

**Summary:**

This work constructs an MMKG for classical Chinese Poetry. This MMKG achieves stronger modality coverage and better quality than previous MMKGs for this purpose. The authors further validate their new MMKG on three different downstream tasks.

**Strengths:**

1. This work has a unique niche, potentially important for the research of classical Chinese Poetry.
2. The construction pipeline is generally well-designed and rigorous.
3. The experimental gain is consistent across tasks and datasets.

**Weaknesses:**

1. (Major) The most fundamental concern is whether multimodal is necessary and beneficial for the research of classical Chinese Poetry. Essentially, poetry is text-only in nature. Therefore, the question can be converted to: whether multimodal is necessary and beneficial for a task that is text-only in nature. There need to be more citations or preliminary studies that can support the claim. However, the first sentence of paragraph 2 has no citations, and the authors simply assert the necessity. The authors could also provide som studies to show that multimodal data can play a necessary role in improving performance on a downstream task other than poetry-image retrieval. Additionally, the benefit of the audio modality is not effectively validated.
2. (Major) The image modality is acquired using a generative approach, which can be concerning. To be specific, (i) there can be noticeable domain gaps between synthetic and real images, and (ii) the model the authors used is relatively outdated (a variant of XL model, released in 2022). Maybe the authors could translate the explanations of PIs into English, so that more recent and stronger models could be used, which do not necessarily understand Chinese.
3. (Minor) The percentage of multimodal data compared to text data appears to be too small.

**Questions:**

See Weaknesses.

---

### Official Review · Reviewer_XJbZ · 2025-11-01

**Soundness:** 2
**Presentation:** 1
**Contribution:** 2
**Rating:** 2
**Confidence:** 3

**Summary:**

This paper introduces CPMK, a large-scale and fine-grained Multimodal Knowledge Graph (MM-KG) designed to advance research on classical Chinese poetry through the integration of textual, visual, and auditory modalities.

CPMK is motivated by the narrow modality coverage, limited scale of existing datasets. To this end, the authors propose a systematic construction pipeline guided by an ontology graph covering multiple aspects of poetry knowledge, resulting in a dataset containing 6.83M textual, 0.21M visual, and 0.08M auditory nodes. Quantitative and qualitative evaluations demonstrate its superior coverage and alignment quality compared to prior datasets (e.g., PKG).

Furthermore, the authors propose KPIR, a Knowledge-enhanced Poetry–Image Retrieval model that leverages modern Chinese translations and poetic imagery knowledge to bridge ancient poetry with corresponding images. In addition, CPMK also enhances performance on poetry theme classification tasks, achieving the best performance on the TCCP dataset.

**Strengths:**

1. Large-Scale and Comprehensive Dataset: The proposed CPMK dataset is the first large-scale multimodal knowledge graph for classical Chinese poetry that integrates textual, visual, and auditory modalities, far exceeding previous resources such as PKG and CCPM in both scale and granularity. The authors justify the necessity of this scale through quantitative comparisons (Table 2) and demonstrate that such comprehensiveness provides a stronger foundation for multimodal poetry research.
2. Empirical Performance Gains in Downstream Tasks: Leveraging CPMK, the authors achieve state-of-the-art results on two downstream applications—poetry–image retrieval and poetry theme classification—surpassing prior baselines. In particular, the proposed Knowledge-enhanced Poetry–Image Retrieval (KPIR) model outperforms strong CLIP-based models across both manually curated and generatively produced datasets, and integrating CPMK into DeepSeek-Chat boosts classification F1 scores on the TCCP dataset to new records.

**Weaknesses:**

1. Limited Technical Innovation and Weak Task Motivation. The core contribution of this work lies in the construction of the CPMK dataset, which integrates textual, visual, and auditory modalities. However, this part is not clearly described in the paper (see below). Moreover, the justification for the downstream tasks is not fully convincing: both poetry–image retrieval and poetry theme classification are well-studied tasks, and the paper does not clearly explain why these tasks are critical for advancing multimodal understanding in classical poetry. In contrast, more creative or generative directions (e.g., poetry creation or cross-modal reasoning) could be more meaningful for modern AI research.
2. Excessive Use of Abbreviations and Concept Overload. In lines 47–52, the paper introduces several abbreviations (AP, PI, IM, II, MCT, etc.) within a short span, many of which are non-essential, making the paper difficult to read smoothly. Furthermore, the conceptual boundaries among these terms are not clearly differentiated, which may confuse readers unfamiliar with the domain. The presentation would be more coherent if only key terms such as CPMK and KPIR were abbreviated, while others were written out in full.
3. Unclear Data Collection and Generation Pipeline. Although Section 3 describes a multimodal data construction process combining web crawling (e.g., SouYun, HanDian) and image generation using diffusion models (e.g., Taiyi-1B), the specific pipeline remains ambiguous. The statistics of the portion of web crawled images versus the generated images, how text and audio data is obtained and refined, are not clearly described.

**Questions:**

Please refer to weakness.

---

### Official Review · Reviewer_Vgza · 2025-11-02

**Soundness:** 3
**Presentation:** 2
**Contribution:** 2
**Rating:** 4
**Confidence:** 4

**Summary:**

This paper proposes CPMK, a large-scale multimodal knowledge graph for classical Chinese poetry, combining textual, visual, and auditory modalities. The authors design an ontology for poetry-related entities, employ knowledge augmentation to improve textual coverage, use LLMs for prompt optimization to improve image generation, and apply CLIP-based alignment to select high-quality text-image pairs. The dataset is partly evaluated both qualitatively (human judgment) and quantitatively (CLIPScore), and its utility is tested on three downstream tasks: poetry–image retrieval, question answering, and theme classification. The authors claim state-of-the-art results and plan to open-source CPMK.

**Strengths:**

1. The work explores a culturally rich and under-studied domain, i.e., classical Chinese poetry, through multimodal data integration. The proposal to replace manual or web-scraped imagery with LLM-optimized generative image synthesis is intuitive and addresses the shortage of multimodal knowledge graph data in this field.

2. The dataset is massive in scale and aims to serve as a foundational resource for a specific cultural domain. The inclusion of downstream experiments provides some evidence of usefulness.

**Weaknesses:**

1. Scientific contribution:

This paper introduces a dataset and benchmark built by assembling well-established components: web crawlers, basic deduplication, LLM rewriting, and standard image generation models. It offers minimal methodological innovation. The key contributions of "knowledge augmentation," "prompt optimization," and "text-image alignment" are incremental applications of existing tools, not foundational advances. Furthermore, these claimed contributions do not tackle a recognized, significant challenge in the field.

2. Weak validation experiments:

- I am concerned that the scale of the poetry-image retrieval benchmark is too small to support reliable conclusions. With only 70 manual and 500 generated pairs, the dataset lacks the statistical power to meaningfully validate the model's performance, making the experimental results potentially unreliable.

- Additionally, since KPIR was trained on the authors' own dataset, its outperformance against other baselines is natural. This makes it hard to conclude that the data or method is uniquely effective. A more convincing validation would be to train models on different components of the data to prove the high quality of each part.

- The superior performance on the theme classification task appears to rely primarily on the capability of DeepSeek-Chat. To fairly validate the effectiveness differences of the PKG and CPMK, I recommend applying them to other baseline models, including more proprietary models, to demonstrate that the gains are generalizable and not model-specific.

3. Poor quality assurance of generated modalities:

The visual and auditory components lack rigorous validation. Images are generated from T2I models with vague “prompt optimization,” yet the paper provides no expert or large-scale human verification of cultural authenticity. For auditory data, there is no evaluation of tone accuracy or phonetic correctness. This undermines the claim of “fine-grained, high-quality multimodal alignment.”

4. Poor writing and presentation:

The paper's positioning of CPMK as a "foundation" is somehow overstated. The current evidence, limited to niche cultural tasks, lacks the generality required to support such a broad claim. Its central contribution is obscured by excessive details of the implementation. The argumentation would benefit from greater focus, shifting from describing the data pipeline to critically analyzing its impact. A conceptual figure illustrating the pipeline would greatly aid clarity.

**Questions:**

Please see weaknesses. I have some additional questions as follows:

- Given that the dataset includes an audio modality, it is unclear why no corresponding downstream tasks were applied to validate its quality, as was done for the image and text modalities. This omission represents a missed opportunity to comprehensively evaluate the dataset's multi-modal claims.

- The readability of the manuscript is significantly hampered by the excessive use of abbreviations. They are defined early in the intro, but their constant recurrence throughout the text forces the reader to continually refer back, creating a disruptive reading experience.

---

### Official Review · Reviewer_gJbF · 2025-11-04

**Soundness:** 3
**Presentation:** 3
**Contribution:** 3
**Rating:** 4
**Confidence:** 3

**Summary:**

This paper points out that research on classical Chinese poetry is limited by existing datasets with single modality, small scale, and insufficient granularity. To address these issues, the authors propose a comprehensive method for constructing the CPMK (Classical Chinese Poetry Multimodal Knowledge Graph). CPMK is a large-scale, fine-grained knowledge graph that integrates three modalities: text, vision, and audio.
The core of the CPMK construction method includes three key aspects: 1) Designing a detailed ontology graph to guide the construction of CPMK. 2) Adopting a knowledge enhancement strategy to integrate and deduplicate textual knowledge from multiple sources. 3) Using an image generation model (Taiyi-1B) to synthesize "Image of Imagery (II)" corresponding to "Poetic Imagery (PI)". To ensure the quality of generated images, the authors use LLMs to optimize image generation prompts, and then apply an image-text alignment strategy based on image-text retrieval and distractors to filter out low-quality image-text pairs.
Through quantitative and qualitative evaluations, as well as comprehensive assessments of three downstream tasks (poetry-image retrieval, poetry question answering, and poetry theme classification), the authors demonstrate the practical value of CPMK.

**Strengths:**

1. The paper specifically elaborates on the significant flaws of existing resources (especially PKG) in Section 1, such as the lack of Poetic Imagery (PI) knowledge, invalid Image of Imagery (II) links, and confusing, noisy Imagery Meaning (IM). These issues provide sufficient justification for the construction of CPMK.

2. The focus and highlight of this paper lie in how to construct the visual modality information of classical poetry.The authors recognize that the "Raw Imagery Meaning (Raw IM)" crawled from the internet is of low quality (overly long or noisy) for image generation models. They use LLMs to convert it into concise visual descriptions suitable for generation.In the further filtering of image-text pairs, the authors do not rely on a simple CLIP Score threshold. Instead, they design a more robust, retrieval-based filtering method (by introducing text distractors). This further ensures the high relevance of the final image-text pairs.

3. The paper makes multiple contributions. It not only provides a high-quality knowledge graph (CPMK) but also contributes two new benchmark datasets for poetry-image retrieval, promoting the development of this field.

**Weaknesses:**

1. The paper emphasizes in the abstract, introduction (and Table 2) that CPMK is a "text, vision, and audio" trimodal knowledge graph. Regrettably, however, the audio modality information is not evaluated or used throughout the entire paper. No downstream tasks utilize this data, nor is any evaluation conducted on its quality. This makes the claim of being "trimodal" somewhat empty, as its value remains unproven.

2. The entire visual modality information is generated by a single model (Taiyi-1B), which may introduce strong model bias. When downstream models learn from CPMK, they may acquire the "visual representation of Poetic Imagery (PI) in the eyes of Taiyi-1B" rather than the "universal visual representation of the Poetic Imagery (PI)".

3. Although the paper demonstrates the relative advantages of CPMK over PKG in quantitative and qualitative analyses in Section 4, the evaluation of downstream tasks is insufficient to fully prove CPMK’s superiority.
In the poetry question answering task in the appendix, the authors mention that ChatGPT-4-RAG (with CPMK applied to RAG) shows significant performance in most tasks. Table 6 presents the task metrics of ChatGPT-4-RAG compared to other base models. However, it is common for RAG to enhance the capabilities of base models, and it remains unclear whether the improvement in task metrics stems from the application of RAG or the unique enhancement ability of CPMK.
In Section 5.1, the authors propose a knowledge-enhanced retrieval model KPIR, which can fully utilize various relevant knowledge in CPMK (PI, IM, II, MCT). This raises a question: In the subsequent poetry-image retrieval task, is the main contributor to achieving SOTA (Table 4) the novel KPIR model architecture or the high-quality CPMK data?
In the poetry theme classification task in Section 5.2, the authors use PI in poetry as queries to retrieve relevant IM from CPMK and PKG. They then combine this knowledge with the original poetry and input it into DeepSeek-Chat for poetry theme classification. The experimental results show that DeepSeek-Chat + CPMK achieves the highest task metrics. However, the gap between this result and those of DeepSeek-Chat alone and DeepSeek-Chat + PKG is small. This outcome may weaken the understanding of CPMK’s superiority.

**Questions:**

1. (Regarding Weakness 1) The audio modality is an important feature of CPMK, but it has not been verified or used in the paper. Could you comment on the intended use of this part of the data? How is the quality of this data verified?
2. (Regarding Weakness 2) Have you evaluated the use of multiple different generation models to generate images? Or could you fully explain the potential impact of using a single model for image generation? Are there methods to mitigate this bias, or is mitigation necessary?
3. (Regarding Weakness 3) For the poetry question answering task, could you explain the unique enhancement ability of CPMK? For example, supplement experiments on ChatGPT-4-RAG with PKG applied to compare its performance with ChatGPT-4-RAG with CPMK in this task.For the poetry-image retrieval task in Section 5.1, could you supplement experiments using other knowledge graphs on different models to make the comparative experiments more comprehensive, thereby demonstrating the high quality of CPMK data?

---

### Note · Authors · 2025-11-20

I have read and agree with the venue's withdrawal policy on behalf of myself and my co-authors.